# Generating Molecules via Chemical Reactions

**John Bradshaw**
University of Cambridge
Max Planck Institute, Tübingen
jab255@cam.ac.uk

**Matt J. Kusner**
University of Oxford
Alan Turing Institute
mkusner@turing.ac.uk

**Brooks Paige**
Alan Turing Institute
University of Cambridge
bpaige@turing.ac.uk

**Marwin H. S. Segler**
BenevolentAI
marwin.segler@benevolent.ai

**José Miguel Hernández-Lobato**
University of Cambridge
Microsoft Research Cambridge
Alan Turing Institute
jmh233@cam.ac.uk

## Abstract

Over the last few years exciting work in deep generative models has produced models able to suggest new organic molecules by generating strings, trees, and graphs representing their structure. While such models are able to generate molecules with desirable properties, their utility in practice is limited due to the difficulty in knowing how to synthesize these molecules. We therefore propose a new molecule generation model, mirroring a more realistic real-world process, where reactants are selected and combined to form more complex molecules. More specifically, our generative model proposes a bag of initial reactants (selected from a pool of commercially-available molecules) and uses a reaction model to predict how they react together to generate new molecules. Modeling the entire process of constructing a molecule during generation offers a number of advantages. First, we show that such a model has the ability to generate a wide, diverse set of valid and unique molecules due to the useful inductive biases of modeling reactions. Second, modeling synthesis routes rather than final molecules offers practical advantages to chemists who are not only interested in new molecules but also suggestions on stable and safe synthetic routes. Third, we demonstrate the capabilities of our model to also solve one-step retrosynthesis problems, predicting a set of reactants that can produce a target product.

## 1 Introduction

The ability of machine learning to generate structured objects has progressed dramatically in the last few years. One particularly successful example of this is the flurry of developments devoted to generating realistic molecules (Gómez-Bombarelli et al., 2018; Segler et al., 2017; Kusner et al., 2017; Dai et al., 2018; Simonovsky and Komodakis, 2018; De Cao and Kipf, 2018; Jin et al., 2018; Liu et al., 2018; You et al., 2018). These models have been shown to be extremely effective at finding molecules with desirable properties: drug-like molecules (Gómez-Bombarelli et al., 2018), biological target activity (Segler et al., 2017), and soluble molecules (De Cao and Kipf, 2018).

However, these improvements in molecule searching come at a cost: these methods do not describe how to synthesize such molecules, a prerequisite for experimental testing. While traditional methods such as virtual screening (Shoichet, 2004) do provide this information, these methods are based on taking a fixed set of molecular fragments and enumerating their combinations using hand-crafted bonding rules. Thus they create very conservative sets of molecules, often with poorer values of properties of interest due to a lack of exploration and focus.

In this paper we propose a generative model (Figure 1) that is able to generate molecules, while describing how to make such molecules from a set of commonly-available reactant molecules. Our model first generates a set of reactant molecules, and then maps them to a predicted product molecule

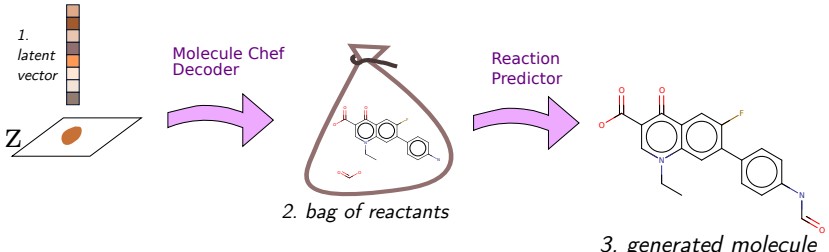

Figure 1: We approach the generation of molecules through a multistage process mirroring how complex molecules are created in practice, with suitable reactants first being found which react together to create a final molecule.

via a reaction prediction model. This mimics closely the real-world process of designing new molecules. Our model also allows one to simultaneously search for better molecules and describe how such molecules can be made.

We show that our model: 1. Improves extrapolation away from the training data; 2. Addresses practical synthesis concerns such as reaction stability and toxicity; and 3. Allows us to propose new reactants for given target molecules that may be more practical to manage.

## 2 BACKGROUND

We start with an overview of traditional computational techniques to discover novel molecules with desirable properties. We then review recent work in machine learning (ML) that seeks to automate parts of this process. We then point out aspects of molecule discovery we believe deserve much more attention from the ML community. We end by laying out our contributions to address these concerns.

### 2.1 VIRTUAL SCREENING

To discover new molecules with certain properties, one popular technique that is used is called *virtual screening* (Shoichet, 2004; Pyzer-Knapp et al., 2015). Virtual screening works by: (a) constructing a library of molecules by enumerating all combinations of an initial set of building blocks that are merged via virtual chemical reactions, (b) for each molecule calculating properties of interest (or proxies of these properties) via simulations or prediction models, (c) filtering the most interesting molecules to synthesize in the lab.

While virtual screening is general, it has an important downside: The number of molecules that can be practically enumerated and screened ($\in [10^7, 10^{10}]$) is small compared to the combinatorially large space of possible drug-like compounds ($\in [10^{23}, 10^{100}]$) (van Hilten et al., 2019).

### 2.2 THE MOLECULAR SEARCH PROBLEM

To address this problem, one idea is to replace this full enumeration with a search algorithm; an idea called *de novo-design* (Schneider and Schneider, 2016). Specifically, instead of generating a large set of molecules with small variations, try to search for molecules with particular properties, recompute these properties for the newfound molecules, and search again. We call this **the molecular search problem**. Early work on the molecular search problem used genetic algorithms, ant-colony optimization, or other discrete search techniques to make local changes to molecules, or steer enumeration of fragment combinations (Schneider and Schneider, 2016; Schneider, 2013; Hartenfeller and Schneider, 2011). While more directed than library-generation, these approaches still explored locally, limiting the diversity of discovered molecules.

The first work to apply current ML techniques to this problem was Gómez-Bombarelli et al. (2018) (in an earlier 2016 preprint with the same name). Their idea was to search by learning a mapping from molecular space to continuous space and back. With this mapping the discrete search problem is transformed into a continuous search problem. Now it is possible to leverage well-studied optimization techniques to do search: local search can be done via gradient descent and global search via Bayesian

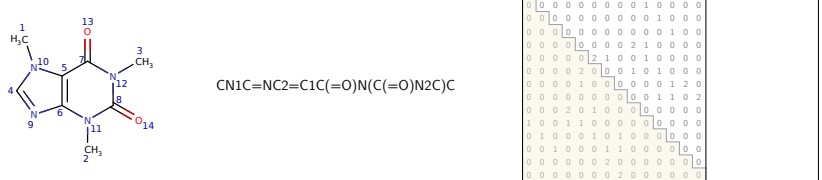

Figure 2: Different approaches for representing the caffeine molecule. From the left we have: (1) a molecular graph representation, the blue digits represent atom mapped numbers given to each atom. (2) A canonical SMILES string representation, (3) The adjacency matrix and node list representation. Note the adjacency matrix is symmetric. The node list may also more information such as charge.

optimization (Snoek et al., 2012; Gardner et al., 2014). For such a mapping, the authors chose to represent molecules as SMILES strings (Weininger, 1988) and leverage advances in generative models for text (Bowman et al., 2016) to learn a character variational autoencoder (CVAE) (Kingma and Welling, 2014). Alongside this work, Segler et al. (2017) trained recurrent neural networks (RNNs) to take properties as input and output SMILES strings with these properties, with molecule search being done using reinforcement learning (RL).

**In Search of Molecular Validity.** However, the SMILES representation is very brittle. Individual atoms and bonds are represented as characters, and if individual characters are changed or swapped, it may no longer represent any molecule (called an *invalid* molecule). Thus, the CVAE, which mapped continuous space to individual SMILES characters, often produced invalid molecules (in one experiment Kusner et al. (2017) sampling from the continuous space produced valid molecules only 0.7% of the time). To address this validity problem Kusner et al. (2017) proposed to instead represent molecules as parse trees from a context-free grammar. They then designed a generative model, called the grammar variational autoencoder (GVAE), to map back and forth from parse trees to continuous space. While this addresses certain problems, there are also non-context-free aspects of SMILES strings that went unmodeled. Dai et al. (2018) therefore extended this approach to attribute grammars which significantly improved the validity. In an independent line of work, work based on RL have sought to directly improve molecular validity during training (Guimaraes et al., 2017; Janz et al., 2018).

Instead of outputting SMILES strings or parse trees, multiple recent works have proposed to map continuous space to a graph representation of molecules (see Figure 2 for an overview of how you can represent a molecule in different ways). Simonovsky and Komodakis (2018); De Cao and Kipf (2018) describe a VAE and a generative adversarial network (GAN) that map directly to the adjacency matrix of a molecular graph. Li et al. (2018) proposed a mapping that builds a molecular graph node-by-node. However none of these approaches solved the validity problem completely. To guarantee molecular validity, Jin et al. (2018); Liu et al. (2018); You et al. (2018) construct molecular graphs iteratively and ensure at each step that the generated graphs are chemically valid.

**Search by Sequential Improvement.** Another parallel line of work called *Matched Molecular Pair Analysis* (MMPA) (Griffen et al., 2011; Dalke et al., 2018) tries to learn an 'improvement function': given an initial molecule, this function outputs an 'improved' one with better properties. It does so by taking a dataset of pairs of molecules, where the second molecule is similar to the first but with better properties. Recently, Jin et al. (2019) have leveraged state-of-the-art advances in machine translation (Zhao et al., 2018) to extend the work of Jin et al. (2018) to learn such an improvement function. They show their method outperforms the works of Jin et al. (2018); You et al. (2018).

### 2.3 THE MOLECULAR RECIPE PROBLEM

Crucially, all of the works in the previous section solving **the molecular search problem** focus purely on optimizing molecules towards desirable properties. They do not address *how practical these molecules are to make*. In practice, the primary limiting factor on making molecules is whether they can be made out of *readily-available starting molecules*. Virtual screening addressed this concern by combining molecular fragments using bonding rules to only generate practical molecules, albeit

with poorer optimization performance. However, in an effort to better solve the molecular search problem, the above works in machine learning tabled these concerns.

To address this concern is to address **the molecular recipe problem**: what molecules are we able to make given a set of readily-available starting molecules? So far this problem has been addressed independently of the molecular search problem through synthesis planning, where the target molecule is recursively deconstructed into increasingly simpler building blocks via formally reversed chemical reactions (Segler et al., 2018). To obtain the molecule in the lab, this plan is executed in the forward direction starting with an initial set of building blocks, similar to a cooking recipe. For this purpose, synthesis planning relies on *reaction predictors*: models that predict how reactant molecules produce a product molecule. More recently, novel ML models have been designed for reaction prediction (Wei et al., 2016; Segler and Waller, 2017; Jin et al., 2017; Schwaller et al., 2018a; Bradshaw et al., 2019; Coley et al., 2019b; Schwaller et al., 2018b). However, even if such methods were used on the molecules found by current generative models, there is no guarantee that the reactants necessary to produce these molecules would be obtainable themselves.

## 2.4 THIS WORK

In this paper, we propose to address both **the molecular search problem** and **the molecular recipe problem** jointly. To do so, we propose a generative model over molecules using a two-step mapping: First, a mapping from continuous space to a set of known, reliable, easy-to-obtain reactant molecules. Second a mapping from this set of reactant molecules to a final product molecule, based on a reaction prediction model (Wei et al., 2016; Segler and Waller, 2017; Jin et al., 2017; Schwaller et al., 2018b; Bradshaw et al., 2019). Thus our generative model not only generates molecules, but also reaction plans that describe *how to generate these molecules from available reactants*. Not only does this address the molecular recipe problem, the learned continuous space can also be used for search. Compared to previous work, here we are searching for new molecules using chemical reactions.

Concretely, we argue that our model has several advantages over the current deep generative models of molecules reviewed in the previous section:

**Better extrapolation properties**  Generating molecules through graph editing operations, representing reactions, gives us strong inductive biases which we hope will allow us to extrapolate further away from any training dataset.

**Validity of generated molecules**  Naive generation of molecular smiles strings or graphs can lead to molecules that are invalid. Although the syntactic validity can be fixed by using masking (Kusner et al., 2017; Liu et al., 2018), the molecules generated can often still be semantically invalid. We hope that instead our model, by generating molecules from chemically stable reactants by means of reactions, proposes more semantically valid molecules.

**Provide synthesis routes**  Proposed molecules from other methods can often not be evaluated in practice, as chemists do not know how to synthesize them. As a byproduct of our model we suggest synthetic routes, which could have useful, practical value.

We introduce our model in the next section, and show in the following section how it addresses both the molecule search and recipe problems.

## 3 MODEL

In this section we describe our model. We define the set of all possible valid molecular graphs as $\mathcal{G}$, with an individual graph $g \in \mathcal{G}$ representing the atoms of a molecule as its nodes, and the type of bonds between these atoms (we consider single, double and triple bonds) as its edge types. The set of common reactant molecules, easily procurable by a chemist, which we want to act as building blocks for any final molecule is a subset of this, $\mathcal{R} \subset \mathcal{G}$.

As discussed in the previous section (and shown in Figure 1) our generative model for molecules consists of the composition of two parts: (1) a decoder from a continuous latent space, $\mathbf{z} \in \mathbb{R}^m$,

to a bag (ie multiset[1]) of easily procurable reactants, $\mathbf{x} \subset \mathcal{R}$; (2) a reaction predictor model that transforms this bag of molecules into a multiset of product molecules $\mathbf{y} \subset \mathcal{G}$.

The benefit of this approach is that for step (2) we can pick from several existing reaction predictor models, including recently proposed methods that have used ML techniques (Kayala et al., 2011; Segler and Waller, 2017; Schwaller et al., 2018b; Bradshaw et al., 2019; Coley et al., 2019a). In this work we use the Molecular Transformer of Schwaller et al. (2018b), as it has recently been shown to provide state-of-the-art performance in this task (Schwaller et al., 2018b, Table 4).

This leaves us with the task of (1), learning a way to decode to (and encode from) a bag of reactants, using a parameterized encoder $q(\mathbf{z}|\mathbf{x})$ and decoder $p(\mathbf{x}|\mathbf{z})$. We call this co-occurance model MOLECULE CHEF, and by moving around in the latent space we can *select* using MOLECULE CHEF different "bags of reactants".

Again there are several viable options of how to learn MOLECULE CHEF. For instance one could choose to use a VAE for this task (Kingma and Welling, 2014; Rezende et al., 2014). However, when paired with a complex decoder these models are often difficult to train (Bowman et al., 2016; Alemi et al., 2018), such that much of the previous work for generating graphs has has tuned down the KL regularization term in these models (Liu et al., 2018; Kusner et al., 2017). We therefore instead propose using the WAE objective (Tolstikhin et al., 2018), which involves minimizing:

$$L = \mathbb{E}_{\mathbf{x} \sim \mathcal{D}} \mathbb{E}_{q(\mathbf{z}|\mathbf{x})} \left[ c(\mathbf{x}, p(\mathbf{x}|\mathbf{z})) \right]$$
$$+ \lambda D \left( \mathbb{E}_{\mathbf{x} \sim \mathcal{D}} \left[ q(\mathbf{z}|\mathbf{x}) \right], p(\mathbf{z}) \right)$$

where $c$ is cost function, that enforces the reconstructed bag to be similar to the encoded one, and $D$ is a divergence measure which forces the marginalised distribution of all encodings to match the prior on the latent space, which is weighted in relative importance by $\lambda$. Following Tolstikhin et al. (2018) we use the maximum mean discrepancy (MMD) divergence measure, with $\lambda = 10$ and a standard normal prior over the latents. We choose $c$ so that this first term matches the reconstruction term we would obtain in a VAE, i.e. with $c(\mathbf{x}, \mathbf{z}) = -\log p(\mathbf{x}|\mathbf{z})$. This means that the objective only differs from a VAE in the second, regularisation term, such that we are not trying to match each encoding to the prior but instead the marginalised distribution over all datapoints. Empirically, we find that this trains well and does not suffer from the same local optimum issues as the VAE.

## 3.1 ENCODER AND DECODER

We can now begin describing the structure of our encoder and decoder. In these functions it is often convenient to work with $n$-dimensional vector embeddings of graphs, $\mathbf{m}_g \in \mathbb{R}^n$. Again we are faced with a series of possible alternative ways to compute these embeddings. For instance, we could ignore the structure of the molecule and learn embeddings for each, or use fixed molecular fingerprints, such as Morgan Fingerprints (Morgan, 1965). We instead choose to use deep graph neural networks (Merkwirth and Lengauer, 2005; Duvenaud et al., 2015; Battaglia et al., 2018) that can produce graph-isomorphic representations of graphs.

Deep graph neural networks have been shown to perform well on a variety of tasks involving small organic molecules, and their advantages compared to the previously mentioned alternative approaches are that (1) they take the structure of the graph into account and (2) they can learn which characteristics are important when forming higher level representations. In particular in this work we use 4 layer Gated Graph Neural Networks (GGNN) (Li et al., 2016). These can compute higher level representations for each node, which are combined by a weighted sum, to form a graph level representation that is invariant to the order of the nodes, in an operation referred to as an aggregation transformation (Johnson, 2017, §3).

**Encoder** The structure of MOLECULE CHEF's encoder, $q(\mathbf{z}|\mathbf{x})$, is shown in Figure 3. For the $i$th data point the embedder is faced with encoding the multi-set of reactants $\mathbf{x_i} = \{x_1^i, x_2^i, \cdots\}$. It first computes the representation of each graph using the GGNN, before summing these representations to get a representation that is invariant to the order of the multiset. A feed forward network is then used to parameterize the mean and variance of a Gaussian distribution over $\mathbf{z}$.

---

[1]Note how we allow molecules to be present multiple times as reactants in our reaction, although practically many reactions only have one instance of a particular reactant.

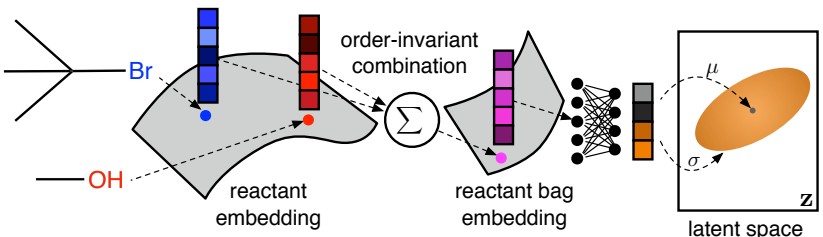

Figure 3: The encoder of our MOLECULE CHEF. This maps from a multiset of reactants to a distribution over latent space. There are three main steps: (1) the reactants molecules are embedded into a continuous space by using Gated Graph Neural Networks (GGNN) (Li et al., 2016) to form molecule embeddings; (2) the molecule embeddings of all the reactants in the multiset are summed to form one order-invariant embedding for the whole multiset; (3) this is then used as input to a single hidden layer neural network which parameterizes an independent Gaussian distribution over each dimension of $\mathbf{z}$.

---

**Algorithm 1** MOLECULE CHEF's Decoder

---

**Require:** $\mathbf{z}^i$ (latent space sample), GGNN (for embedding molecules), RNN (recurrent neural network), $\mathcal{R}$ (pool of easy-to-obtain reactant molecules), $\mathbf{s}$ (learnt "halt" embedding), $\mathbf{A}$ (matrix that projects the size of the latent space to the size of RNN's hidden space)
    $\mathbf{h}_0 \leftarrow \mathbf{A}\mathbf{z}^i$ {The latent input is projected to the same space as the RNN's hidden layer.}
    $\mathbf{m}_0 \leftarrow \mathbf{0}$ {Start symbol}
    **for** $t = 1$ to $T_{\max}$ **do**
        $\mathbf{h}_t \leftarrow \text{RNN}(\mathbf{m}_{t-1}, \mathbf{h}_{t-1})$
        $\mathbf{B} \leftarrow \text{STACK}([\text{GGNN}(g) \text{ for all } g \text{ in } \mathcal{R}] + [\mathbf{s}])$
        $\text{logits} \leftarrow \mathbf{h}_t \mathbf{B}^T$
        $x_t \sim \text{softmax}(\text{logits})$
        **if** $x_t = \text{HALT}$ **then**
            break {If the logit corresponding to the halt embedding is selected then we stop early}
        **else**
            $\mathbf{m}_t \leftarrow \text{GGNN}(x_t)$
        **end if**
    **end for**
    return $x_1, x_2, \cdots$

---

**Decoder**    The decoder, $p(\mathbf{x}|\mathbf{z})$, maps from the latent space to a multiset of reactant molecules. These reactants are typically small molecules, which means we could fit a deep generative model which produces them from scratch. However, to better mimic the process of selecting reactant molecules from an easily obtainable set, we instead restrict the output of the decoder to pick the molecules from a fixed set of reactants molecules, $\mathcal{R}$.

This happens in a sequential process using a recurrent neural network (RNN), with the full process described in Algorithm 1 and illustrated in Figure 4. The latent vector, $\mathbf{z}$ is used to parametrize the initial hidden layer of the RNN. The selected reactants are fed back in as inputs to the RNN at the next generation stage. Whilst training we randomly sample the ordering of the reactants, and use teacher forcing.

### 3.2 ADDING A PREDICTIVE PENALTY LOSS TO THE LATENT SPACE

As discussed in section 2.2 we are interested in using and evaluating our model's performance in the **molecular search problem**, that is using the learnt latent space to find new molecules with desirable properties. In reality we would wish to measure some complex chemical property that can only be measured experimentally. However, as a surrogate for this, following (Gómez-Bombarelli et al., 2018), we optimize instead for the QED score of a molecule, $\mathbf{w}$, as a deterministic mapping from molecules to this score, $\mathbf{y} \mapsto \mathbf{w}$, exists in the popular chemo-informatics toolkit RDKit (RDKit, online). This score gives a Quantitative Estimate of Drug-likeness (Bickerton et al., 2012).

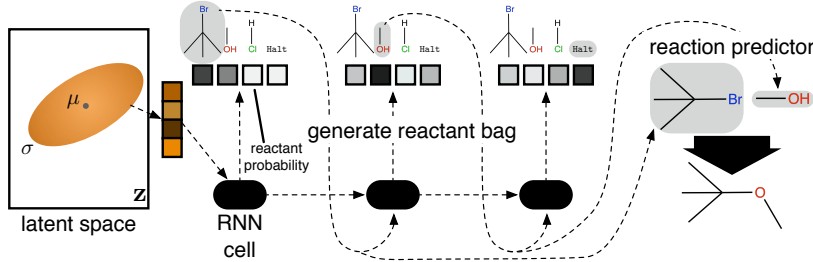

Figure 4: MOLECULE CHEF's decoder. The decoder generates the multiset of reactants in sequence through calls to a RNN. At each step the model picks either one reactant from the pool or to halt, finishing the sequence. The latent vector, $\mathbf{z}$, is used to parameterize the initial hidden layer of the RNN. Reactants that are selected are fed back into the RNN on the next step. The reactant bag formed is later fed through a reaction predictor to form a final product.

To this end, in a similar manner to Liu et al. (2018, §4.3) & Jin et al. (2018, §3.3), we simultaneously train a 2 hidden layer property predictor NN. This network tries to predict the QED property, $\mathbf{w}$, of the final product $\mathbf{y}$ from the latent encoding of the associated bag of reactants. The use of this property predictor network for local optimization is described in Section 4.2.

# 4 EVALUATION

In this section we evaluate MOLECULE CHEF in (1) its ability to generate a diverse set of valid molecules; (2) how useful its learnt latent space is when optimizing product molecules for some property; and (3) whether by training a regressor back from product molecules to the latent space, MOLECULE CHEF can be used as part of a setup to perform retrosynthesis.

In order to train our model we need a dataset of reactant bags. For this we use the USPTO dataset (Lowe, 2012), processed and cleaned up by Jin et al. (2017). We further use a subset of reactions exhibiting linear electron flow topologies (Herges, 1994), so the reactants can interact in simple ways. We filter out reagents, molecules that form context under which the reaction occurs but do not contribute atoms to the final products, by following the approach of Schwaller et al. (2018a, §3.1).

We wish to use as possible reactant molecules only popular molecules that a chemist would have easy access to. To this end, we filter our dataset so that each reaction only contains reactants that occur at least 15 times across different reactions in the original larger dataset. This leaves us with a dataset of 21928 unique reactant bags (and corresponding products). In total there are 3180 unique reactants.

## 4.1 GENERATION

| Model Name | Validity | Uniqueness | Novelty |
|---|---|---|---|
| MOLECULE CHEF + MT | 99.0 | 91.7 | 87.1 |
| LSTM (Segler et al., 2017) | 88.6 | 89.8 | 72.9 |
| CVAE (Gómez-Bombarelli et al., 2018) | 4.3 | 72.7 | 93.5 |
| GVAE (Kusner et al., 2017) | 5.0 | 85.7 | 100.0 |

Table 1: Table showing the validity, uniqueness and novelty (all as %) of the products/or molecules generated from decoding from 20k random samples from the prior $p(\mathbf{z})$. MT stands for the Molecular Transformer (Schwaller et al., 2018b).

We begin by analysing our model using the metrics favored by previous work (Jin et al., 2018; Liu et al., 2018; Li et al., 2018; Kusner et al., 2017). In particular, we look at validity (how many of the molecules generated by the model can be parsed by RDKit), uniqueness (how many different distinct molecules we sample, conditioned on them being valid) and novelty (what percentage of the valid molecules do not appear in the training set).

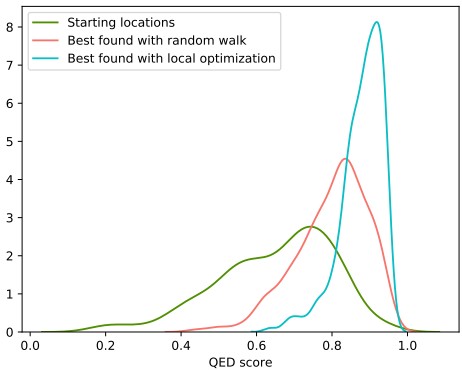

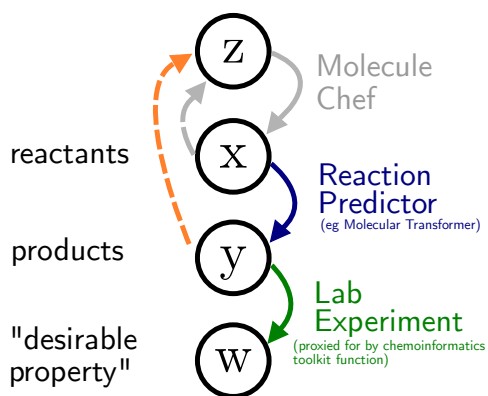

Figure 5: KDE plot showing that the distribution of the best QEDs found through local optimization, using our trained property predictor for QEDs, has higher mass over higher QED scores compared to the best found from a random walk. The starting locations' distribution (sampled from the training data) is shown in green. The final products, given a reactant bag are predicted using the Molecular Transformer (Schwaller et al., 2018b).

Figure 6: Having learnt a latent space which can map to products through reactants, we can learn a regressor back from the suggested products to latent space (orange arrow shown) and couple this with MOLECULE CHEF's decoder to see if we can do retrosynthesis – the act of computing the reactants that create a particular product.

For the baselines we consider the character VAE (Gómez-Bombarelli et al., 2018), the grammar VAE (Kusner et al., 2017), and a stacked LSTM generator with no latent space (Segler et al., 2017). The training set is our 3180 unique reactants, and the associated products from the reactions these partake in (even though MOLECULE CHEF has not seen the latter during training, the reaction predictor has). Note how in total the training set is far smaller in size (<30k molecules) compared to the reaction datasets commonly used in other works assessing generative models of graphs, such as ZINC (Irwin et al., 2012), which has 250k molecules.

Computing the validity, uniqueness and novelty for the output of MOLECULE CHEF coupled with a reaction predictor is not as straightforward. This is because this model can output multiple molecules for one location in latent space, reflecting the fact that chemical reactions can form more than one product. We therefore instead define validity as requiring that at least one of the molecules in the bag of products can be parsed by RDKit. For a bag of products to be unique we require it to have at least one valid molecule that the model has not generated before in any of the previously seen bags. Finally, for computing novelty we require that the valid molecules not be present in the same training set we use for the baseline generative models.

The results are shown in Table 1. As MOLECULE CHEF decodes to a bag made up from a predefined set of molecules, those reactants going into the reaction predictor are valid. The validity of the final product is not 100%, as the reaction predictor can make non-valid edits to these molecules, but we see that in a high number of cases the products are valid too.

## 4.2 LOCAL OPTIMIZATION

As discussed in Section 3.2, when training MOLECULE CHEF we simultaneously trained a property predictor network, that tries to learn a function from the latent space of MOLECULE CHEF to the QED score of the final product. In this section we look at using the gradient information obtainable from this property network to do local optimization to find a molecule created from our reactant pool that has a high QED score.

We evaluate the local optimization of molecular properties by taking 250 bags of reactants, encoding them into the latent space of MOLECULE CHEF, and then repeatedly moving in the latent space using the gradient direction of the property predictor until we have decoded ten different reactant bags. As a comparison we consider instead moving in a random walk until we have also decoded to ten different reaction bags. In Figure 5 we look at the distribution of the best QED score found in considering these ten reactant bags, and how this compared to the QEDs started with.

When looking at individual optimization runs, we see that the QEDs vary a lot between different products even if made with similar reactants. However, Figure 5 shows that overall the distribution of the final best found QED scores is improved when purposefully optimizing for this. This is encouraging as it gives evidence of the utility of these models for the molecular search problem.

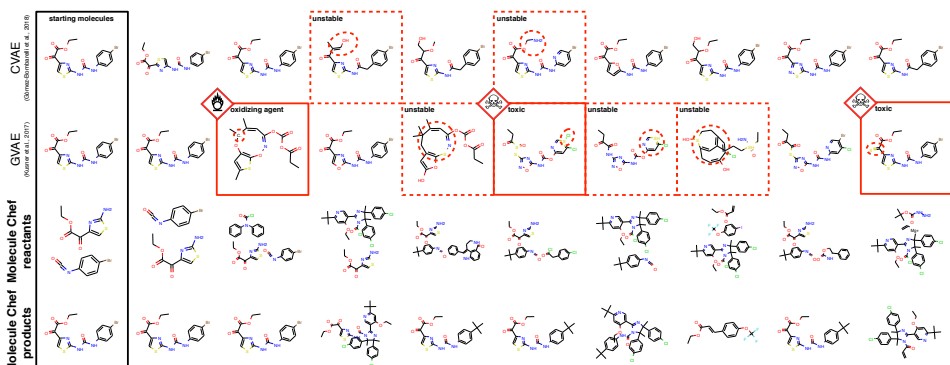

Figure 7: Random walk in latent space. See text for details.

## 4.3 RETROSYNTHESIS

A unique feature of our approach is that we learn a decoder from latent space to a bag of reactants. This gives us the ability to do retrosynthesis by training a model learning a mapping from products to their associated reactants' representation in latent space and using this in addition to MOLECULE CHEF's decoder to generate a bag of reactants. This process is highlighted in Figure 6.

Although retrosynthesis is a difficult task, with often multiple possible ways to create the same product and with current state-of-the-art approaches built using large reaction databases and able to deal with multiple reactions (Segler et al., 2018), we believe that our model could open up new interesting and exciting approaches to this task. We therefore train a small network based on the same graph neural network structure used for MOLECULE CHEF followed by four fully connected layers to regress from products to latent space.

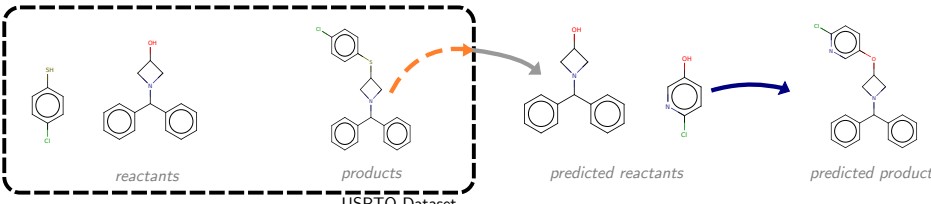

Figure 8: An example of performing one-shot retrosynthesis prediction using a trained regressor from products to latent space. This reactant-product pair has not been seen in the training set of MOLECULE CHEF. Further examples are shown in the appendix.

A few examples of the predicted reactants corresponding to products that MOLECULE CHEF has not seen in training before, but which can be made in one step from the pre-defined possible reactants, are shown in Figure 8 and the appendix. We see that often this approach, although not always able to suggest the correct whole reactant bag, chooses similar reactants that on reaction produce similar structures to the original product we were trying to synthesize. While we would not expect this approach to retrosynthesis to be competitive with complex planning tools, we think this provides a promising new approach, which could be used to identify bags of reactants that produce molecules similar to a desired target molecule. In practice, it would be valuable to be pointed directly to molecules with similar properties to a target molecules if they are easier to make than the target, since it is the properties of the molecules, and not the actual molecules themselves that we are after.

## 4.4 QUALITATIVE QUALITY OF SAMPLES

In Figure 7 we show molecules generated from a random walk starting from the encoding of a particular molecule (shown in the left-most column). We compare the CVAE, GVAE, and MOLECULE

CHEF (for MOLECULE CHEF we encode the reactant bag known to generate the same molecule). We showed all generated molecules to a domain expert and asked them to evaluate their properties in terms of their *stability*, *toxicity*, *oxidizing power*, *corrosiveness*. Many molecules produced by the CVAE and GVAE show undesirable features, unlike the molecules generated by MOLECULE CHEF.

## 5 DISCUSSION

In this work, we have introduced MOLECULE CHEF a model that generates synthesizable molecules. By constructing molecules through selecting reactants and running chemical reactions, while performing optimization in continuous latent space, we can combine the strengths of previous VAE-based models and classical discrete de-novo design algorithms based on virtual reactions.

ACKNOWLEDGEMENTS

This work was supported by The Alan Turing Institute under the EPSRC grant EP/N510129/1. JB also acknowledges support from an EPSRC studentship.

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

# A APPENDIX

## A.1 FURTHER RANDOM WALK EXAMPLE

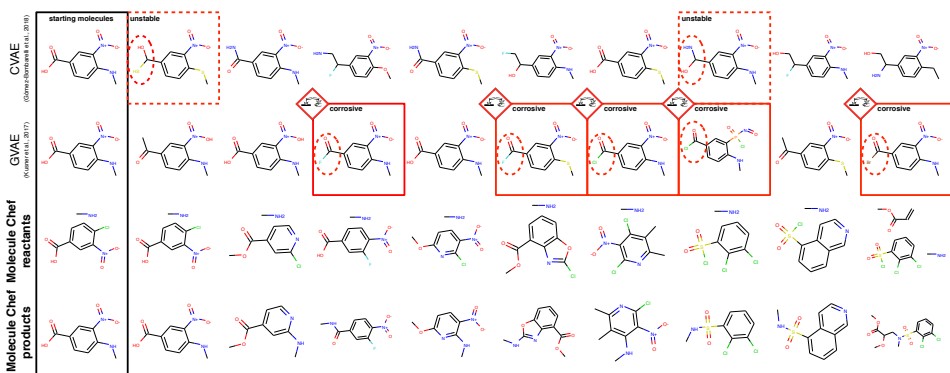

Figure 9: Another example random walk in latent space.

## A.2 FURTHER RETROSYTHESIS EXAMPLES

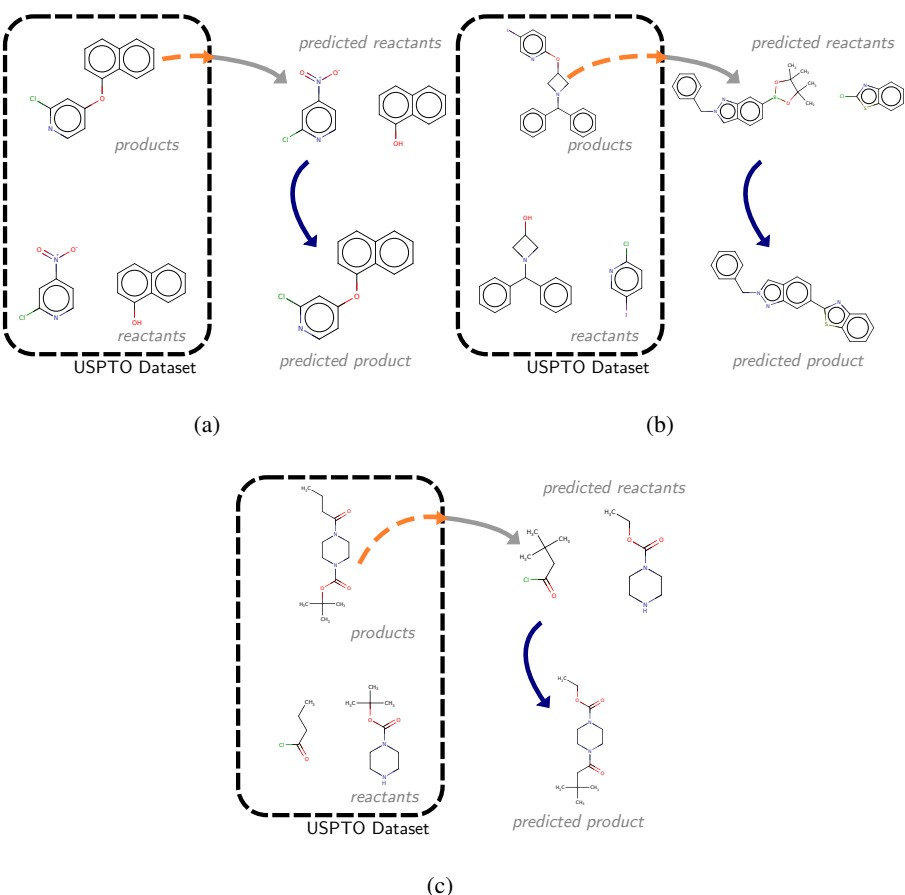

(a)        (b)

(c)

Figure 10: Further examples of the predicted reactants associated with a given product for product molecules not in MOLECULE CHEF's training dataset.

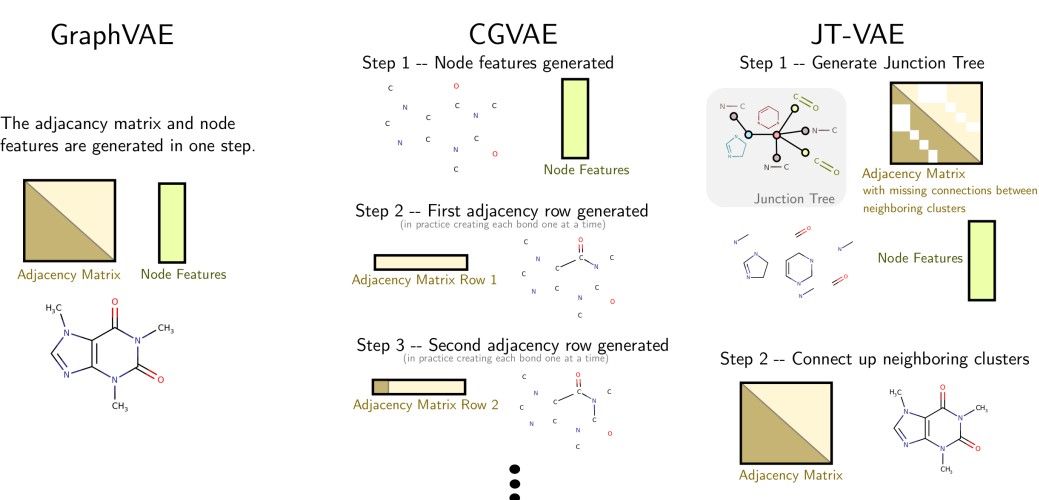

Figure 11: Approaches of previous methods for creating graphs describing molecules. From left to right we have GraphVAE (Simonovsky and Komodakis, 2018), Constrained Graph VAE (CGVAE) (Liu et al., 2018) and the Junction Tree VAE (JT-VAE) (Jin et al., 2018). Note that some of the steps shown in the diagram can be further broken down. For instance the CGVAE creates the adjacency matrix rows in a series of steps, each one creating one non-zero entry. Likewise, the JT-VAE creates the initial tree in an iterative process.

