# OpenReview forum: "Generating Molecules via Chemical Reactions"
_ICLR.cc/2019/Workshop/DeepGenStruct — DeepGenStruct 2019_

### Official Review · AnonReviewer1 · 2019-04-12

**Rating:** 4
**Confidence:** 3

**Review:**

This paper proposes a molecular generative model that generates molecules via a two-step process: 1) Generate a set of reactants from the latent space; 2) Predict the product of the generated reactants using a pre-trained reaction predictor. This two-step formulation provides synthesis routes of generated molecules, allowing end users to examine the synthetic accessibility of generated compounds. This approach is analogous to virtual screening approach in medicinal chemistry.

The proposed approach builds on a Wassarstein autoencoder. The encoder maps a set of reactants into a continuous vector using a gated graph neural network followed by a sum pooling. The decoder is a recurrent network that decodes the reactant molecules one by one. The output of the decoder is restricted to be a fixed set of reactant molecules, and the decoding process is modeled as generating a sequence of tokens (reactant ID).

This paper is well motivated. The reviewer agrees that the molecular recipe problem is important: the model should provide hints (e.g. one-step synthesis routes) of how generated molecules can be synthesized, so that chemists can synthesize the suggested compounds for experimental validation.

My concern of the proposed model is that it can only generate molecules that can be synthesized through a one-step reaction from a fixed reactant vocabulary (3180 reactants). This may imply that the proposed model can only cover a limited subset of chemical space. Another concern is lack of quantitive evaluation: For the local optimization experiment, the paper didn't compare with any previous approach (e.g., CVAE, GVAE). For retrosynthesis experiment, there is no quantitive evaluation at all. One suggestion is to conduct human evaluation: how often do the chemists think the suggested retrosynthesis plans make sense.

Nonetheless, the proposed method is interesting combination of generative modeling and chemical reaction prediction. The reviewer therefore votes for the acceptance of the paper.

---

### Official Review · AnonReviewer2 · 2019-04-15
**Interesting paper with thorough experiments**

**Rating:** 4
**Confidence:** 2

**Review:**

This paper presents Molecule Chef, a model that maps between continuous latent codes and bags of readily available reactants. When combined with existing models that predict what products will result from a reaction with the given reactants, this model can be used to propose methods to synthesize new compounds. The continuous latent space is also useful for enabling continuous search for reactions that lead to products with desired properties.

Overall, this is a strong paper, both in terms of motivation and results:
- The pipeline of first sampling multisets of reactants from Molecule Chef, then using an existing model to predict products, generates 99% valid molecules (molecules known to exist) with a high degree of novelty (molecules not in the training data). This compares favorably with comparable baselines from previous work.
- It's possible to learn a predictor from latent code directly to some desirable property of the end product, such as QED score. Using this, it is possible to search in the latent space for reactions that lead to products with high QED score. It is shown that gradient-based search (which can only be done in this latent space, and not in the reactants space) is better than a naive random walk. This justifies the use of a continuous latent code, instead of directly generating bags of reactants.

Some questions I had:
- It would be nice to know how well a model that doesn't have the latent code, and instead directly generates bags of reactants, does in terms of the generation quality metrics in Section 4.1, and the retrosynthesis experiments in 4.3.
- Section 4.4 was unclear--what are the "undesirable features", and why would we expect Molecule Chef to avoid such features?

---

### Decision · Program_Chairs · 2019-04-19
**Acceptance Decision**

Accept